# Children’s Internet Use Profiles in Relation to Behavioral Problems in Lithuania, Latvia, and Taiwan

**DOI:** 10.3390/ijerph17228490

**Published:** 2020-11-16

**Authors:** Vilmantė Pakalniškienė, Roma Jusienė, Sandra B. Sebre, Jennifer Chun-Li Wu, Ilona Laurinaitytė

**Affiliations:** 1 Institute of Psychology, Faculty of Philosophy, Vilnius University, LT-01513 Vilnius, Lithuania; roma.jusiene@fsf.vu.lt (R.J.); ilona.laurinaityte@fsf.vu.lt (I.L.); 2 Department of Psychology, Faculty of Education, Psychology and Art, University of Latvia, LV-1083 Riga, Latvia; sandra_beatrice.sebre@lu.lv; 3 Department of Early Childhood and Family Education, College of Education, National Taipei University of Education, Taipei 10671, Taiwan; jenwu@tea.ntue.edu.tw

**Keywords:** internet use, school-aged children, latent profiles, SDQ

## Abstract

This study explored the profiles of elementary-school-aged children’s Internet use in relation to their emotional and behavioral problems. Participating in this cross-sectional study were 877 child–parent dyads from Latvia, Lithuania, and Taiwan. Children (8–10 years old) provided information on three variables: the amount of time they spent online, frequency of online activities, and knowledge of how to do things online. Latent profile analysis including these three variables provided a four-class solution for child Internet use. A comparison between Latvia, Lithuania, and Taiwan on the percentage of the sample distribution in each class showed that there was no difference between sites for the high class (high ratings on all three variables). The largest differences were for the low and average classes (low and average ratings on all three variables, namely, time online, frequency, and knowledge): the Lithuanian and Taiwanese samples were similar in that a higher percentage of each sample was in the low class, whereas the Latvian sample had children equally distributed between the low class and the average class. Analysis of the data from the entire sample for differences in parent-reported child behavioral difficulties suggested that children in the high class had an elevated level of behavioral problems and compulsive Internet use.

## 1. Introduction

The Internet has become an important part of our daily lives, and both children and adults are experiencing the ubiquitous nature of the Internet. Various forms of digital technology surround children in their home, school, or public places, and they are using the Internet with increasing frequency at very young ages (starting in infancy) [1]. At present in many countries, it is almost imperative that parents buy home computers or other digital devices and subscribe to the Internet in order to provide their children with tools that are necessary for educational engagement. However, children often engage with digital devices for other types of activity as well, such as social networking and playing video games. Thus, the increasing amount of time that children are engaged with digital devices raises questions about the impact of these activities on their psychological well-being.

Researchers have proposed that Internet use be considered on a continuum ranging from healthy use to excessive or problematic use, and that problematic use is linked to the frequency and type of the online interaction for adolescents and young adults [2,3]. Educators and other specialists assert that the judicious use of the Internet provides new opportunities for learning, creativity, and communication, thereby promoting language, cognitive, and social development [1]. Studies with adolescents show that the benefits provided by Internet use include the improvement of various skills, such as communication, social relations, and competence in using digital technologies [4]. In a systematic review of studies with adolescents and young adults, researchers found that the identified benefits of online communication included enhanced self-esteem, greater social support from peers, and an increased ability to disclose their feelings [5]. Studies have also provided evidence for the positive effects of Internet use on academic accomplishments. In a study with low-income children and adolescents (10–18 years old) who were provided with Internet access by the researchers [6], the frequency of Internet use was associated with higher scores on standardized tests. In another study, at-home online learning (excluding playing or browsing) was related to better development of expressive language and metacognitive planning for children aged 6–12 years old [7].

The potentially negative impact of excessive Internet use has been studied both in relation to the frequency of global or overall Internet use, as well as in relation to the frequency of specific forms of Internet use. Previous studies have shown that augmented computer use may decrease the time for social engagement with others in face-to-face interaction and could affect children’s social development [8]. An increase in social isolation may foster feelings of depression and loneliness, and it may also affect language development and academic achievement for children and adolescents [9,10,11,12]. Screen time similarly displaces time for other elementary school-age children’s activities that are known to enhance cognitive development, such as socializing, physical activity, or reading a book [13]. A greater frequency of screen time has been associated with various markers of poor physical and psychosocial health, including adiposity, lower self-esteem, and lower academic achievement, as well as indicators of higher depression and anxiety for adolescents [14].

Internet use research has accentuated the differentiated effects that are linked to different types of screen use. Results from a meta-analysis indicated a negative association between the frequency of social networking and academic performance for adolescents and young adults [15], and a recent study by Walsh and colleagues [16] showed that a higher overall screen time spent watching TV or videos or viewing social media was associated with lower cognitive test scores for children aged 9–10 years old. Männikkö and colleagues [17] provided a systematic review and meta-analysis of problematic gaming behavior and found that moderate amounts of gaming can serve as a means of relaxation and stress reduction, but that excessive gaming is associated with various detrimental health-related outcomes, such as depression and anxiety. Finally, exposure to aggressive or violent content while using a computer has been related to aggressive behavior offline for early adolescents [18].

Among the various detrimental mental-health-related outcomes of excessive Internet use is compulsive Internet use (CIU), which is marked by difficulty in controlling one’s use of the Internet, the need for increasingly greater time spent using the Internet, and withdrawal symptoms, such as irritability when one is not using the Internet [19]. Adolescent preoccupation with Internet use has been related to various mental health problems, such as depression and anxiety, as well as problems in relationships [20]. The sequelae of CIU for adolescents has been differentiated by types of Internet use, with CIU symptoms being facilitated by the frequency of gaming, surfing, downloading, chatting, and messaging, but not with the frequency of seeking information [21]. Similarly, Blinka and colleagues [22] found particularly detrimental mental health symptoms from online gaming for study participants aged 11–16 years old, and the study by Gamito and colleagues [23] showed a greater risk of Internet use preoccupation for adolescents who engage in more active Internet-related activities, such as online gaming and the production of multimedia for the purposes of online uploading.

Finally, active engagement in online activities demands a certain amount of relevant skills and knowledge, and since Internet use has become such an inevitable aspect of modern life, the enhancement of one’s digital literacy has become an important issue for the well-being of young people [24]. Digital literacy is essential such that one may fully benefit from the advantages provided by digital technology. Generally, the more adolescents use the Internet, the more digital literacy skills they have; yet, at the same time, they not only have more opportunities to engage in various types of online activities but also to face various online risks [25].

In reviewing the existing research on Internet use, it is apparent that the majority of studies have been with adolescents and emerging adults, whereby children have been relatively left behind. Livingstone and Haddon [26] have argued that since digital media use amongst children is increasing so rapidly, and since children are more vulnerable to potential risks due to their lack of maturity and less developed coping strategies, it is imperative for researchers to place greater emphasis on the study of children younger than 12 years old in relation to their media use. Consequently, this study aimed to evaluate the association between Internet use and the emotional and behavioral problems of elementary school children between the ages of 8 and 10.

The present study was based upon an international collaborative project that was investigating elementary-school-aged children’s Internet use in relation to various opportunities, risk factors, and detrimental mental health problems. For the purposes of cross-cultural comparison, samples were provided from Taiwan, Lithuania, and Latvia, regions that are similar in regard to children’s widespread use of digital technology but differ in regard to cultural traditions, family values, and other aspects of sociocultural context. Researchers have often explored sociocultural differences along the individualism–collectivism continuum, and studies on cultural dimensions indicate that Taiwan ranks low on individualism, whereas Latvia and Lithuania are moderately individualistic, with Latvians being somewhat more individualistic than Lithuanians [27]. Contemporary household Internet access in Latvia, Lithuania, and Taiwan is fairly comparable, with the present-day Internet availability at nearly total population coverage, especially in regard to households with children. At the beginning of 2018 in Latvia, 81.6% of households had Internet access, an increase of 21.8 percentage points since 2010 [28]. Statistics in Lithuania [29] from the first quarter of 2018 showed that 78% had Internet access at home, which is 3 percentage points more than in 2017. In Taiwan, household access to the Internet increased from 70.6% in 2005 to 84.9% in 2018 [30,31]. Internet access is available to 98.1% of households with children in Latvia [28] and available to 97% of households with children in Lithuania [29]. Similar statistics are available from various Asian countries, for example, in Singapore, 98% of households with school-aged children have Internet access [32].

In this study, we aimed to identify the profiles of children according to the frequency of their online activities (frequency), their general knowledge of how to do perform digital operations (knowledge), and the amount of time spent online (time). We assumed that children who do not have enough knowledge of digital operations but spend a lot of time online could experience various risks more often than others, and this could be associated with emotional and behavioral problems and CIU. The research questions of this study were as follows: Can we identify different Internet use classes by frequency, knowledge, and time in a sample of children aged 8 to 10 years old? Does the proportion of children in each class differ in Lithuania, Latvia, and Taiwan? Are there differences in the ratings of emotional symptoms, behavioral problems, and CIU between children from different classes? In this study, we hypothesized that there would be several classes that would differ in their Internet use profile according to frequency, knowledge, and time. We also hypothesized that there would be differences by class in parent-reported emotional and behavioral problems and CIU.

## 2. Materials and Methods

### 2.1. Participants

The participants of this study, both children and their parents, were recruited as part of a larger international collaborative project between Latvia, Lithuania, and Taiwan. The larger study included data collection taken one year apart at time 1 and time 2 (the falls of 2018 and 2019). This paper was based upon the time 1 data only. The time 1 sample included 877 children aged between 8 and 10 years (48.3% female): 269 children from Latvia, 304 from Lithuania, and 304 from Taiwan. Parents provided permission for their child to participate in the study, and one of the child’s parents was also requested to complete a set of questionnaires, with 863 parents returning completed questionnaire forms. Of these parents, 719 (82%) were the child’s biological mother, 124 (14.1%) were the child’s biological father, and the remaining 20 (2.3%) were either step-parents, foster parents, or grandparents. A more detailed description of the time 1 study participants by data collection site is presented in Table 1.

In Riga (Latvia), 389 parents received information and an invitation to participate in the study, with 269 parents providing written consent for their child’s and their own participation (response rate of 69.2%). In Vilnius (Lithuania), 450 consent forms with information about the study were distributed to parents, resulting in 293 parents who provided written consent (response rate of 65.1%). However, some of the parent questionnaires (*n* = 26) were only partially completed and therefore not included in the data analysis. In Taipei (Taiwan), there were 442 parents invited to participate in the study. A total of 304 parents who agreed to participate in this study received a parent-report questionnaire to complete at home, and their child completed a questionnaire during the group administration at school. This resulted in a response rate of 68.8%. 

### 2.2. Procedure

Prior to the onset of the study, the researchers in Latvia, Lithuania, and Taiwan received approval for the data collection from their relevant ethics committee: from the University of Latvia in Latvia (07.11.18. V69/15), Vilnius University in Lithuania (2018-10-22, no.18), and National Taiwan University in Taiwan (201705ES030). Data collection was carried out by the researchers and their research assistants. In accord with a previous agreement between all three research teams, they contacted several typical public schools from a major city in their region, excluding schools with a specialized focus. With permission from the school’s administration, parents/caregivers of children in the second and third grades (in the third and fourth grades in Taiwan for comparable ages of 8–10 years) were invited to participate in the study. All of the parents or caregivers of children in the classes chosen for participation were sent consent forms, where there were no exclusion criteria. Parents were informed about confidentiality and the purpose of the study, and upon their agreement, they were asked to provide written consent for their child’s and their own participation in the study. Children were also informed about the purpose of the study, the principles of confidentiality, and about their right to refuse participation in the study, even if their parents had provided written consent. All of the children with written consent from their parents provided active consent for their own participation. The children completed the paper-and-pencil questionnaire in their classroom during their regular school hours, with a research assistant assisting in the process.

### 2.3. Measures

#### 2.3.1. Time Spent Online

In order to assess the amount of time spent online, the children were asked to respond to the following two questions: “About how long do you spend on the Internet per day during a regular weekday (school day)? About how long do you spend on the Internet per day during a weekend or holiday?” For each question, the child was asked to indicate the amount of time spent online from choices on a nine-point scale, ranging from (1) little or no time to (9) 7 h or more. Data from these two questions (online time during weekdays and online time during weekends) was averaged for the final analysis because the correlation between these two items was found to be moderately positive (*r* = 0.57, *p* < 0.00) for the entire sample. The means and standard deviations for the sample were as follows: M = 2.40 (SD = 1.77) and M = 3.57 (SD = 2.32) for time spent online during weekdays and weekends, respectively. The means and standard deviations of these items for Latvia, Lithuania, and Taiwan are presented in Table 2.

#### 2.3.2. Frequency of Online Activities

The children were presented with a list of twelve common online activities: using the Internet for homework, searching for information, finding out how much something costs, talking to people, sending and receiving messages, social networking, playing online games, listening to music online, watching video clips, watching TV shows or movies, creating video or music and uploading, taking photos, and creating images and sharing. Children were asked to respond to the following question “How often have you done these things online in the past month?” by rating each type of online activity on a five-point scale ranging from (1) never or hardly never to (5) several times each day. The child’s ratings for each of these twelve activities were averaged in order to obtain the child’s overall frequency of online activities. Higher scores indicated more frequent Internet use. The Cronbach’s alpha for this scale for the entire sample was 0.81. The mean and standard deviation for the entire sample was M = 2.28, SD = 0.71. The reliability estimates, means, and standard deviations of the scale for Latvia, Lithuania, and Taiwan are presented in Table 2.

#### 2.3.3. Knowledge of How to Do Things Online

Children were presented with a list of twelve common digital technology procedures: knowing how to save a photo or picture that is found online, changing privacy settings, knowing which information should or should not be shared, removing people from contacts, creating videos online, installing apps, making an in-app purchase, opening downloaded files, using shortcut keys, opening a new tab in a browser, finding a website that had been visited before, and creating a website. Children were asked to respond to the following question “How much do you know about doing these things online?” by rating each type of digital procedure on a three-point scale ranging from (1) know nothing about that to (3) know a lot about that. The child’s ratings for these twelve digital technology procedures were averaged in order to obtain a rating of the child’s overall knowledge of how to do things online. Higher scores indicated a higher level of knowledge of performing digital technology procedures. The Cronbach’s alpha for this scale for the entire sample was 0.88. The mean and standard deviation for the entire sample was M = 1.81, SD = 0.54. The reliability estimates, means, and standard deviations of the scales for Latvia, Lithuania, and Taiwan are presented in Table 2.

#### 2.3.4. Children’s Emotional and Behavioral Problems

In order to obtain an evaluation of the child’s emotional and behavioral problems, the child’s parent was asked to complete the 25-item Strengths and Difficulties Questionnaire [33]. Each item is rated on a three-point scale ranging from (0) not correct to (2) absolutely correct. This measure includes five subscales: conduct problems, hyperactivity/inattention, emotional symptoms, peer relationship problems, and prosocial behavior. The Cronbach’s alphas for these subscales for the entire sample were 0.55 for conduct problems, 0.68 for hyperactivity/inattention, 0.63 for emotional symptoms, 0.52 for peer relationship problems, and 0.69 for prosocial behavior. The reliability estimates of these subscales for Latvia, Lithuania, and Taiwan are presented in Table 2.

#### 2.3.5. Compulsive Internet Use

The child’s parents provided an evaluation of their child’s level of preoccupation with online activity by completing the 14-item Compulsive Internet Use questionnaire [19]. The items include symptoms of addiction, such as loss of control, withdrawal, mood modification, preoccupation, and conflict. Items were rated on a five-point scale ranging from (1) never to (5) very often. A mean score for the CIU scale was computed for each participant, with higher scores indicating greater compulsive use of the Internet. The Cronbach’s alpha for this scale for the entire sample was 0.92. The reliability estimates of these subscales for Latvia, Lithuania, and Taiwan are presented in Table 2.

#### 2.3.6. Socioeconomic Status

The child’s parents provided information about their perception of the family’s financial situation. Parents were asked to rate their family’s financial situation by marking one of the following statements: (1) we can afford all that we would like to have, (2) we are fairly well off, (3) we can get along well enough, (4) we have only the very basic necessities, and (5) we lack even the basic necessities. This item is an adaptation of the measure of subjective socioeconomic status developed by Goodman and colleagues [34], with the reliability being demonstrated by intraclass correlations and correlations with objective measures of social status [35].

### 2.4. Data Analysis

IBM SPSS 25 (IBM, Chicago, IL, USA) and Mplus 8.0 (Muthen & Muthen, Los Angeles, CA, USA) [36] were used to carry out the data analysis. Missing or incomplete data is a problem in many studies [37]. Missing data in this study was controlled for by using the full information maximum likelihood (FIML) [38]. The proportion of missing data was calculated with a covariance “coverage” matrix [36] provided by the program. The minimum coverage for the trusted data analyses was 0.10. In this study, the coverage indexes ranged from 0.94 to 0.98.

We performed mixture modeling to identify whether the child’s gender and data collection site (Latvia, Lithuania, or Taiwan) were related to their patterns of Internet use (time spent online, frequency of online activities, and knowledge of how to do things online) and whether these patterns were related to emotional and behavioral problems and CIU. First, we performed a latent class analysis to identify the patterns of Internet use predicted from the continuous measures of time (average time spent online per day), frequency (degree of frequency of activities online), and knowledge (level of knowledge of how to do things online) [39]. Following the presumption that covariates may improve the accuracy of the assignment into certain classes or groups [40], the data collection site and gender were included as covariates in the model (A in Figure 1).

In addition to the initial latent class model, we tested mixture models (as shown in Figure 1) in which the child’s gender and data collection site served as covariates; emotional and behavioral problems, as well as CIU, served as distal outcomes. The mixture models combined several analyses in the same model. Latent class or latent profile analyses are comparable to cluster analysis. Models with distal outcomes estimate the effects of latent class or profile membership on distal outcomes. This provides estimates of the distal outcomes as a comparison between classes [41]. The mixture model indicated whether children from certain classes had more problems than children in the other classes (B in Figure 1). In order to test whether the means of the outcome measures were significantly different between groups/classes, we ran the models with and without holding the means equal and evaluated the chi-square (twice the log-likelihood) differences between classes. The paths between the child’s gender and data collection site and the distal outcomes showed the extent to which gender or the data collection site were related to the level of problems, after controlling for class membership (C in Figure 1). However, this was not the main interest of this study, and therefore, this path was simply controlled for in all of the models. All of these paths in the mixture models were estimated simultaneously.

## 3. Results

### 3.1. Identification of the Latent Classes Based on Internet Use—Frequency, Knowledge, and Time

In order to test the model presented in Figure 1, we ran the latent class model using the children’s reported online activity with frequency, knowledge, and time as variables. Different models with increasing numbers of classes were compared with each other (see Table 3).

The low values of the Bayesian information criterion (BIC), Akaike information criterion (AIC), and sample-size-adjusted Bayesian information criterion (SSABIC), and the high values of the entropy criterion and Lo–Mendell–Rubin adjusted likelihood ratio test suggested that a four-class model represented the best solution for the children’s online behavior. Thus, the final four-class model was used for further analysis as the best estimate of the children’s online behavior (see Figure 2). In order to describe the classes, we designated the terms low, average, or high Internet use if both the frequency and time online variables were low, average, or high, respectively.

The largest observed class, the low class, included 51.7% of the entire sample (*n* = 453) in this model. As shown in Figure 2, the low knowledge and low Internet use (Internet use designates both frequency and time online) class was characterized by low levels of frequency, knowledge, and time (M = −0.87, SE = 0.11 for frequency; M = −60, SE = 0.12 for knowledge; and M = −1.15, SE = 0.16 for time spent online).

The second class, the average knowledge and average Internet use class, accounted for 22.7% of the sample (*n* = 199) in the model. The average class had average levels of frequency (M = 0.88, SE = 0.18) and knowledge (M = 0.69, SE = 0.19) and lower levels of time spent online (M = −0.05, SE = 0.20).

The third class, which was designated the low knowledge and high time online class, included children who were spending quite a bit of time online. This class had lower than average levels of frequency (M = 0.37, SE = 0.20) and knowledge (M = 0.01, SE = 0.21) but higher than average levels of time spent online (M = 2.75, SE = 0.32). This class included 13.6% of the sample (*n* = 119). These children seemed to have a restricted focus on the types of online activities they engaged in and less knowledge of digital operations, but they were spending a lot of time online.

The final class, the high knowledge and high Internet use class, accounted for 8.8% of the sample (*n* = 77) in the model. This class had the highest levels of frequency (M = 2.60, SE = 0.46), knowledge (M = 1.76, SE = 0.43), and time spent online (M = 4.74, SE = 0.59).

We looked at the frequencies of the different types of activity that the children were engaged in online for each class separately. We found that watching video clips was the predominant activity reported by children in all of the classes, where 63 to 65% of children in each class reported engagement with this type of activity. In all of the classes, another dominant activity was gaming online, where 48 to 59% of children in each class reported engagement with gaming. Watching movies ranged from 42 to 67%. Children in the low knowledge–high time online class reported significantly more time watching video clips or movies, gaming, and listening to music, rather than spending time online for schoolwork or finding information for their own interest in comparison to the other classes. Children in the high knowledge and high Internet use class reported spending time watching clips, watching movies, or gaming, but also more than half of the sample reported using the Internet for schoolwork or finding information for their own interest. Children in the low knowledge and low Internet use class reported spending time watching clips, watching movies, and gaming, but also more than half of the sample in this class reported use of the Internet for finding information for their own interest and chatting with others. Children in the average knowledge and average Internet use class reported spending time watching clips and gaming, but also using the Internet for finding information for their own interest and schoolwork, and chatting with others. Even though some online activities, such as watching clips and gaming, were mentioned by children in each of the four classes, there were differences between classes with respect to the types of online activity they engaged in.

Data collection site effect: For evaluating the effect of the data collection site, we used the data collection site as a covariate. We wanted to examine whether living in a certain region would increase or decrease the chance of being in a certain class. In this comparison, the low knowledge and low Internet use class served as the baseline or reference category. The results are presented as odds ratios. The results suggested that children from Taiwan had a lower probability of being in the high knowledge and high Internet use class or in the average knowledge and average Internet use class in comparison to the low knowledge and low Internet use class (OR = −0.55, *p* = 0.02 and OR = −0.53, *p* = 0.01 for the high knowledge and high Internet use class and the average knowledge and average Internet use class, respectively) compared to children from Latvia and Lithuania. It seemed that being a child from Taiwan increased the chance of being in the low knowledge and low Internet use class in comparison to being a child from Latvia or Lithuania.

We used the chi-square difference test in order to evaluate the proportion of children from each site in each class. The results, as seen in Table 4, show that there were significant differences between the proportions of children in each class according to the data collection site (*χ*^2^ = 63.24, df = 6, *p* < 0.001).

We see that there were no significant differences between sites in the proportions of children in the high knowledge and high Internet use class. However, the biggest differences were for the low knowledge and low Internet use class and average knowledge and average Internet use classes. A relatively smaller proportion of children from Latvia were in the low knowledge and low Internet use class, in contrast to the largest proportion of children from Taiwan being in the low knowledge and low Internet use class. For the average knowledge and average Internet use class, it was the other way around, i.e., the smallest proportion of children from Taiwan and the biggest proportion of children from Latvia. In the low knowledge–high time online class, the proportions of Lithuanian and Taiwanese children were similar, but they differed from the proportion of Latvian children. Latvia had the highest proportion of children in this class in comparison to Lithuania and Taiwan. Thus, these differences were similar to results from the logistic regression, where it seemed that being from Taiwan increased the child’s chance of being in a low Internet use class.

Gender effect: We used the child’s gender as a covariate in order to examine to what extent a child’s gender would increase or decrease the chance of them being in a certain class. We once again chose the low knowledge and low Internet use class as the baseline or reference class for the other classes. The results indicated a gender effect (OR = 1.27, *p* = 0.01; OR = 0.32, *p* = 0.21; OR = 0.63, *p* = 0.02 for the high knowledge and high Internet use class, average knowledge and average Internet use class, and low knowledge and low Internet use class, respectively). Results from the mixture model suggested that gender (being a boy) increased the chance of being in a high time online class in comparison to girls. We also used the chi-square difference test in order to evaluate the proportions of boys and girls in each profile. In contrast to the results from the mixture model, the results from the chi-square analysis, as shown in Table 5, suggested that there were no significant differences (however, there was a tendency) between the proportions of boys and girls in each class (*χ*^2^ = 7.29, df = 3, *p* = 0.063).

Even though the child’s gender increased their chance of being in a certain class, we found no gender differences when comparing the proportions of boys and girls in each class.

SES and child’s age effects: SES was not included as a covariate in the analysis because this item showed low correlations with frequency, knowledge, and time online (*r* = 0.08, *p* = 0.03; *r* = 0.03, *p* = 0.37; *r* = 0.10, *p* = 0.02 for frequency, knowledge, and time online, respectively). We used the chi-square difference test in order to evaluate the proportions of different perceptions of the family’s financial situation in each profile. The results indicated no significant differences between the proportions of the perceived family’s financial situation in each class (*χ*^2^ = 8.05, df = 9, *p* = 0.53). Thus, the results showed that the perceived family’s financial situation was not related to the child’s class. Similarly, the child’s age was not included as a covariate because this item had low correlations with frequency, knowledge, and time online (*r* = −0.09, *p* = 0.02; *r* = 0.15, *p* = 0.01; *r* = −0.08, *p* = 0.03 for frequency, knowledge, and time online, respectively). We used the chi-square difference test in order to evaluate the proportions of children at each age in each profile. The results showed no significant differences between the proportions of children at each age in each class (*χ*^2^ = 6.97, df = 6, *p* = 0.32).

### 3.2. Relationships between Internet Use Profiles and Children’s Emotional and Behavioral Problems

To determine whether Internet use was linked to emotional and behavioral problems in the current sample, we examined the links between class membership and the distal outcomes, namely, conduct problems, hyperactivity/inattention, emotional symptoms, peer problems, prosocial behavior, and compulsive Internet use (B in Figure 1). Thus, we tested four separate models with one of the distal outcomes each time. The fit indices for all the models for the entire sample are shown in Table 6.

To examine whether the means of the emotional symptoms, conduct problems, hyperactivity/inattention, peer problems, prosocial behavior, and compulsive Internet use (CIU) outcomes were significantly different between classes, we evaluated the models with and without setting up the means as equal and used the chi-square difference test to compare the two models (one with means set up as equal and one without), as it is suggested for structural equation modeling [42]. As seen in Table 7, all or some of the classes differed from each other when evaluating behavioral problems and when evaluating CIU.

However, the analysis showed no significant differences between the classes for emotional symptoms, peer problems, or prosocial behavior. We did expect that children who were actively spending a lot of time online but did not have enough knowledge (the low knowledge–high time online class children) would be more at risk and have higher levels of behavioral difficulty than the others. However, the results showed that these children were similar to children in the high knowledge and high Internet use class in the degree of their conduct problems and hyperactivity/inattention, and that both of these classes differed from the low knowledge and low Internet use class, which was marked by the lowest levels of conduct problems and hyperactivity/inattention. It seemed that children in the classes with increased online time may be more likely to have conduct problems or to be more hyperactive/inattentive. In an examination of CIU by class, we found that the high knowledge and high Internet use class had the highest levels in comparison to the other classes, while the low class had the lowest level of CIU. The average knowledge and average Internet use class and the low knowledge–high time online class showed similar levels of CIU. Even though the amount of time spent online was found to be related to conduct problems or hyperactivity/inattention, the results indicated that the frequency of Internet activity and increased knowledge of how to do things online was also related to CIU.

## 4. Discussion

Previous studies, primarily with adolescents or emerging adults, have shown that Internet use is related to both risks and negative consequences, as well as opportunities and positive consequences. In the present study with elementary school-age children, we explored these possibilities by identifying separate classes or subgroups of children according to the characteristics of their Internet use, namely, the frequency of online activity, time spent online, and knowledge of how to do things online. We also examined whether the classes would differ according to the degree of emotional or behavioral problems and CIU. We assumed that CIU could be a consequence of certain activities and time spent online, and thus, it was chosen as one of the distal outcomes related to mental health problems. Therefore, one of the unique features of this study was the person-centered approach, by which we identified children with different Internet use behavior profiles.

In this study, we found that extensive and active use of the Internet (e.g., frequency of online activity, having a lot of digitally-related knowledge, and spending quite a bit of time online) was related to a higher likelihood of having conduct problems, hyperactivity/inattention, or CIU. Our results showed that 8-to-10-year-old children who were spending a lot of time online were at risk for more behavioral difficulties than the other children, as indicated by the higher parent-reported ratings for conduct problems and hyperactivity for children in the low knowledge–high time online class and the high knowledge and high Internet use class. Considering that children from the low knowledge–high time online class did not indicate a high frequency of online activity, it may be that the type of activity online is an important factor.

There may be several explanations for the relationships between Internet use and behavioral difficulties. First, it could be that children who do have behavioral problems, especially problems related to impulsivity and dysregulation, are more prone to engage in online activities and to have greater difficulty in stopping themselves from using the Internet. Furthermore, frequent Internet use is likely to negatively affect the child’s social contact, especially their contact with family members and peers [43]. There is the possibility that as a result, their social competence would decline. Thus, decreased social interaction could affect social development and contribute to emotional or behavioral problems in childhood [44,45]. Further on in life, the degree of social competence achieved in childhood could affect their social adjustment in adolescence and adulthood, including the presence or absence of psychopathology [46]. Therefore, frequent or excessive use of digital technology may serve to promote or accentuate the child’s social difficulties, which then makes adjustment more problematic in various aspects of the child’s life. Yet, it should also be pointed out that the results of our study did not indicate any differences in peer problems or prosocial behavior according to the child’s Internet use class. Nevertheless, the results of our study support previous scientific considerations that Internet use may be riskier for specific groups of children [24], i.e., those children having regulatory problems and/or lacking supportive relationships with parents.

Another possible explanation for the link between the increased amount of time spent online, conduct problems, and hyperactivity may be related to the type of online activity the child pursues. According to Kalmus and colleagues [24], children with more psychological difficulties are more prone to risky online activities. In our study, we found that the most popular activities in this sample of 8-to-10-year-old children were playing games and watching video clips. These activities were especially prominent for the high knowledge and high Internet use class and the low knowledge–high time online class children, which were the classes whose parents reported higher levels of conduct problems and hyperactivity/inattention. It is known that the content of both video clips and games often involves aggressive or violent competition. Previous studies have indicated that violent video games have a direct effect on aggression and problem behavior [47,48]. Therefore, repeated exposure and engagement with aggressive or violent online games may be reflected in an increased level of behavioral difficulty. In addition, it may be that video games with increased aggression may be more addictive and therefore have more detrimental outcomes to mental health [22,23,49,50]. The relationship between the specific content and the specific contexts or purposes for which children are using the Internet and their behavioral difficulties require greater attention and should be further addressed in future studies.

In this study, we also looked for differences in emotional symptom levels between classes but we did not find such differences. Previous studies with adolescents and young adults have shown that there are associations between excessive use of the Internet and emotional problems, such as a depressed mood, anxiety, or loneliness [51,52,53]. On the other hand, there are inconsistencies between studies. In one study, the authors found that loneliness was not related to either the amount of time spent online or with e-mail correspondence, but rather it was related to the participants’ gender [54]. Keeping in mind that emotional problems tend to increase as a child enters adolescence [55], it could be that the lack of relationship between Internet use class and emotional symptoms in our study was due to the younger age of our study participants and that the relationship between emotional problems and Internet use will become more evident at a later developmental stage. Finally, at younger ages, Internet use is typically controlled and monitored by parents to a greater extent than it is for adolescents. Therefore, in future studies, it would be worthwhile to consider the possible mediating role of parental involvement and the effect of developmental maturation from a longitudinal perspective.

The results of studies on gender effects in relation to excessive Internet use are contradictory, ranging from no differences at all [56], to the finding that the female gender in adolescence is a significant risk factor for CIU [21], and finally, to the majority of studies and systemic reviews that claim problematic Internet use to be associated with the male gender, both among adolescents and young adults [2,50,57]. The results of our study showed that being a boy increased the chances of being in either the high knowledge and high Internet use class or the low knowledge–high time online class of Internet users. Children in the high knowledge and high Internet use class also reported higher ratings of CIU. Thus, the results of our study implied that it could be important to consider the male gender as a risk factor for problematic Internet use as early as during the preadolescent period.

In this study, we also investigated whether the sociocultural context in which the child was living would increase the child’s chance of being in a certain class. Children from Latvia, Lithuania, and Taiwan participated in our study. Latvia and Lithuania are neighboring Baltic countries, and thus we expected the results from these countries to be fairly similar. In addition, we expected that the results from Taiwan would be significantly different, taking into account the assumption that smartphones would be more accessible to children in Taiwan, as it is in other parts of Asia due to their low price and more advanced telecommunication infrastructures and development [58]. Although similar profiles on Internet use, time spent online, and digital skills were indeed found, interestingly, being a Taiwanese child increased the chance of being in the low knowledge and low Internet use class. On the other hand, the largest proportions of Latvian children were in the average knowledge and average Internet use class and the low knowledge–high time online class. In general, the Internet proved to be available to a high extent in Latvia, Lithuania, and Taiwan; therefore, these differences in Internet use should be further explored in relation to differences in the cultural and educational contexts, as well as in relation to various parental monitoring practices and the implementation of rules regulating children’s Internet use at home and at school. Of note is also the fact that although children in all three samples indicated Internet availability in their homes, there was a marked difference in the degree of Internet accessibility for each individual child. The parents from our sample reported that in Taiwan, only 14.1% of the children had their own personal smartphones at the time of our data collection, while in Latvia and Lithuania, 86 and 80% of children owned their own smartphones, respectively. Having one’s own personal smartphone, by definition, implies the possibility of greater accessibility to the Internet. The reason or rationale for this discrepancy in having one’s own personal smartphone also requires attention in our future studies.

Previous studies have suggested that excessive Internet use may promote various mental health problems, including problematic Internet use [52,53]. In this study, we examined CIU as one of several mental health problems linked to Internet use, in parallel with emotional symptoms and conduct problems. We aimed to explore whether different Internet use patterns (e.g., time spent online, frequency of activities online, and knowledge of how to do things online) would be related to CIU problems in children aged between 8 to 10 years. Our results showed that being active online and spending greater amounts of time online, regardless of the child’s self-reported knowledge and skill level regarding Internet use, was putting the child at risk of developing CIU. Thus the results of our study offer support and justification for the recommendation for caregivers and healthcare providers to attend to the amount of time children are spending online.

This study had several limitations. Our study had a cross-sectional design. This allowed us to identify associations between various factors; however, causal relationships could not be tested. It may be that consistent and frequent use of the Internet promotes behavioral problems, but it may also be that children with certain behavioral difficulties are more willing or interested in using the Internet. Furthermore, previous studies have shown that possible bidirectional relations between a child’s emotional or behavior problems and Internet or media use should be acknowledged [21,24,59]. Thus, longitudinal data could evaluate the directions of effect between Internet use and the child’s adjustment. In this study, we did not confirm the content of the child’s Internet use. Taking into account the fact that previous studies have indicated that violent video games have a direct effect on aggression and problem behavior [47,48], future studies should also evaluate the content of Internet activity. Another consideration is that the children participating in this study were attending school in a major urban center; thus, it should be taken into account that their parents may have had a relatively high socioeconomic status and that, therefore, in our data analysis, we did not find any differences between classes in relation to the parents’ perception of their family’s financial situation. It is a limitation of our study that we assessed only the parents’ subjective perception of their financial situation and did so with only one question. Answers to this question may have been affected by social desirability. In our study, perception of the family’s financial situation was weakly related to Internet use, and therefore, this item was not entered into the model. Future studies should include a more diverse sample in regard to socioeconomic status because with higher diversity in SES, there may be more marked differences in Internet use and also behavioral problems.

Besides these limitations, our results suggested that it is possible to identify elementary school-age children’s Internet use profiles according to the frequency of their Internet use, their knowledge of Internet-related procedures, and the amount of time they spend online. Our findings suggest that children with different Internet use profiles have differences in their degree of behavioral difficulties and Internet preoccupation. This implies that particular attention should be paid to children’s routinized and frequent use of the Internet. Adults caring for a child (e.g., parents and formal and informal educators) should be paying considerable attention to how much time elementary-school-aged children in their care are spending online. They should also beware of the child’s activities when online. It seems that certain rules in the family and at school must be applied in regard to children’s Internet use. Taking into consideration the fact that children with regulatory problems could also be more vulnerable to excessive Internet use and risky activities when online, they should receive consistent, coherent, and rational guidance. The comparison between Lithuania, Latvia, and Taiwan indicated that the child’s actual Internet use accessibility is of importance. Although at the time of our data collection, a great majority of the children from our samples in Latvia and Lithuania already had their own personal smartphone, and since it would not be feasible for parents to remove the smartphones, it is advisable for parents to pay much closer attention to the amount of time that the child spends engaged with the Internet. Looking toward the future, it may be useful to encourage society at large to disparage, rather than encourage, personal smartphones for children in the early elementary school grades.

## 5. Conclusions

It is possible to identify elementary-school-aged children’s Internet use profiles according to the frequency of their Internet use, their knowledge of Internet-related procedures, and the amount of time they spend online. The results suggested that children with different Internet use profiles have differences in their degree of behavioral difficulties and compulsive Internet use. We found that extensive and active use of the Internet (e.g., frequency of online activity, having considerable digitally-related knowledge, and spending quite a bit of time online) was associated with a higher likelihood of having conduct problems, hyperactivity, or CIU. Our results showed that 8-to-10-year-old children who were spending a lot of time online were at risk of displaying more behavioral difficulties than the other children.

## Figures and Tables

**Figure 1 ijerph-17-08490-f001:**
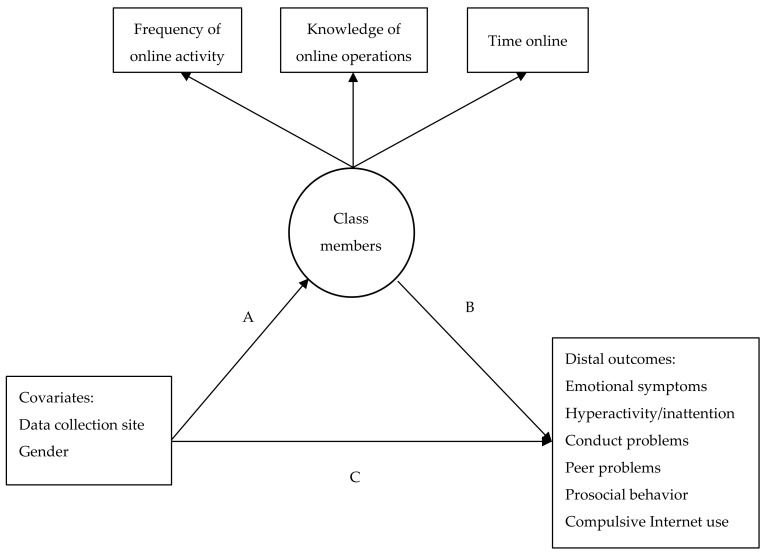
Conceptual model of the latent classes of children’s Internet use with the covariates and outcomes. Note: A—path from covariates to latent classes; B—path from latent classes to distal outcomes; C—path from covariates to distal outcomes.

**Figure 2 ijerph-17-08490-f002:**
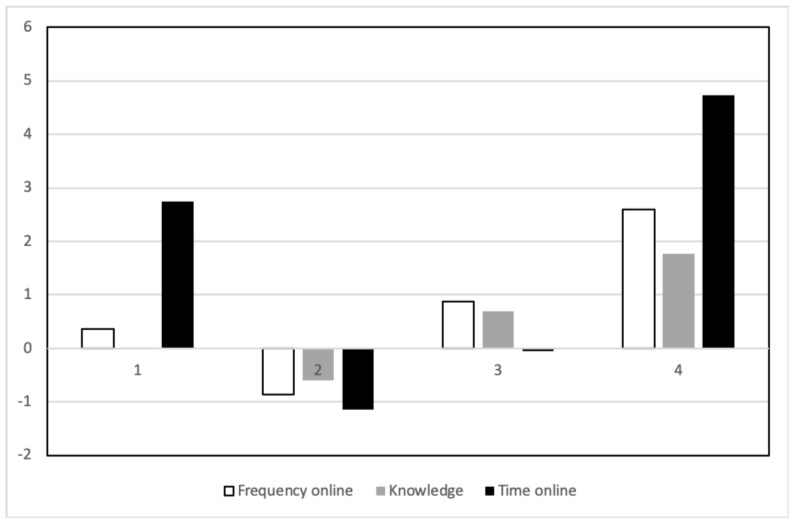
Model-estimated means (expressed in z-scores) for the four-class solution. On the y-axis, the z-scores of the Internet activities variables are given, and on the x-axis, the four profile groups are presented. 1—low knowledge and high time online class, 2—low knowledge and low Internet use (time online and frequency) class, 3—average knowledge and average Internet use (time online and frequency) class, and 4—high knowledge and high Internet use (time online and frequency) class.

**Table 1 ijerph-17-08490-t001:** Characteristics of the participants.

Category	Latvia	Lithuania	Taiwan
	Number of Participants (%)	Number of Participants (%)	Number of Participants (%)
Gender			
Boys	140 (52.0)	144 (47.4)	140 (46.1)
Girls	127 (47.2)	153 (50.3)	164 (53.6)
Age			
8	120 (44.6)	138 (45.4)	38 (12.5)
9	126 (46.8)	129 (42.4)	144 (47.4)
10	13 (4.9)	5 (1.7)	117 (38.5)
Parents’ marital status			
Married	158 (58.7)	242 (80.3)	278 (91.4)
Lives with a partner	58 (21.6)	15 (5.0)	3 (1.0)
Divorced	31 (11.5)	8 (2.7)	18 (6.0)
Other	9 (3.4)	25 (8.2)	4 (1.3)
Employment status			
Working full time	211 (78.4)	236 (77.6)	193 (63.5)
Working part time	13 (4.8)	23 (7.6)	32 (10.5)
Childcare leave	19 (7.1)	20 (6.6)	1 (0.3)
Unemployed	17 (6.3)	7 (2.3)	67 (22.0)
Other	5 (1.9)	7 (2.3)	10 (3.3)

**Table 2 ijerph-17-08490-t002:** Means and standard deviations of the variables used in the data analysis.

Variables	Latvia	Lithuania	Taiwan
Time spent on the Internet			
Working days, mean (SD)	2.85 (1.81)	2.47 (1.82)	1.93 (1.56)
Weekends, mean (SD)	4.08 (2.20)	3.39 (2.28)	3.38 (2.41)
Frequency of online activities			
Scale, mean (SD)	2.55 (0.73)	2.36 (0.71)	1.98 (0.59)
Cronbach’s alpha	0.80	0.77	0.80
Knowledge			
Scale, mean (SD)	1.93 (0.51)	1.80 (0.56)	1.71 (0.54)
Cronbach’s alpha	0.86	0.88	0.91
Children’s emotional and behavioral problems			
Scale, mean (SD)			
Conduct problems	5.89 (1.53)	5.71 (1.23)	5.51 (1.49)
Hyperactivity/inattention			
Emotional symptoms	7.22 (1.82)	7.19 (1.87)	6.86 (1.78)
Peer problems			
Prosocial behavior	12.94 (1.89)	12.83 (1.80)	12.26 (1.92)
Cronbach’s alpha			
Conduct problems	0.58	0.53	0.55
Hyperactivity/inattention	0.70	0.66	0.70
Emotional symptoms	0.61	0.65	0.65
Peer problems	0.49	0.36	0.56
Prosocial behavior	0.69	0.66	0.71
Compulsive Internet use			
Scale, mean (SD)	2.55 (0.68)	2.43 (0.74)	1.89 (0.67)
Cronbach’s alpha	0.90	0.93	0.90

**Table 3 ijerph-17-08490-t003:** Model indices for different latent class models.

Number of Latent Classes	Number of Parameters	Log-Likelihood	BIC	SSABIC	AIC	LMR Adj. LRT	Entropy
1 class	6	−3524.56	7089.58	7070.53	7061.12	-	-
2 classes	12	−3290.05	6660.91	6620.80	6604.11	391.80 *p* = 0.07	0.81
3 classes	18	−3211.28	6543.77	6486.61	6458.57	153.74 *p* = 0.01	0.68
4 classes	24	−3151.83	6465.26	6385.04	6351.66	95.41 *p* = 0.001	0.75
5 classes	30	−3102.94	6407.88	6312.61	6265.88	116.04 *p* = 0.08	0.75

Note: BIC, Bayesian information criterion; SSABIC, sample-size-adjusted Bayesian information criterion; AIC, Akaike information criterion; LMR Adj. LRT, Lo–Mendell–Rubin adjusted likelihood ratio test.

**Table 4 ijerph-17-08490-t004:** Number of children from different data collection sites in each class.

Class	Latvia, *N* (%)	Lithuania, *N* (%)	Taiwan, *N* (%)
Low Knowledge and Low Internet Use	100 (37.2)	162 (59.1)	210 (69.1)
Average Knowledge and Average Internet Use	107 (39.8)	68 (24.8)	52 (17.1)
Low Knowledge and High Time Online	44 (16.4)	29 (10.6)	32 (10.5)
High Knowledge and High Internet Use	18 (6.7)	15 (5.5)	10 (3.3)

**Table 5 ijerph-17-08490-t005:** Number of boys and girls in each class.

Class	Girls, *N* (%)	Boys, *N* (%)
Low Knowledge and Low Internet Use	232 (56.7)	214 (49.8)
Average Knowledge and Average Internet Use	92 (22.5)	107 (24.9)
Low Knowledge and High Time Online	58 (14.2)	61 (14.2)
High Knowledge and High Internet Use	27 (6.6)	48 (11.2)

**Table 6 ijerph-17-08490-t006:** The fit indices for the mixture models.

Distal Outcome/Fit Indices	Emotional Symptoms	Hyperactivity/Inattention	Conduct Problems	Peer Problems	Prosocial Behavior	Compulsive Internet Use
Log-likelihood	−4878.81	−4953.02	−4662.67	−4825.29	−4908.63	−4024.21
BIC	9953.77	10,102.19	9521.50	9846.40	10,013.42	8243.73
SSABIC	9861.68	10,010.09	9429.40	9754.64	9921.32	8151.63
AIC	9815.62	9964.04	9383.35	9708.59	9875.26	8106.42
Entropy	0.73	0.73	0.73	0.73	0.73	0.75
Number of parameters	29	29	29	29	29	29

**Table 7 ijerph-17-08490-t007:** Estimates of the various problems by class in the mixture models with emotional and behavioral problems and compulsive Internet use as distal outcomes.

Distal Outcomes	Low Knowledge and Low Internet Use Class	Average Knowledge and Average Internet Use Class	Low Knowledge and High Time Online Class	High Knowledge and High Internet Use Class
Emotional symptoms	3.85	3.84	4.04	4.06
Hyperactivity/inattention	2.87 _a_	3.10	3.15 _b_	3.18 _b_
Conduct problems	3.89 _a_	4.15	4.23 _b_	4.27 _b_
Peer problems	3.46	3.53	3.60	3.40
Prosocial behavior	6.65	6.75	6.81	6.83
Compulsive Internet use	1.78 _a_	2.57 _b_	2.63 _b_	4.29 _c_

Note: Within each row, means with different letters differ significantly between classes at the *p* < 0.05 level. a—the lowest mean that significantly differ from the average (b) and the highest (c) value; b—the average mean value that significantly differ from the lowest (a) and the highest (c) value; c—the highest mean value that significantly differ from the lowest (a) and/or the average (b) value. The results were obtained while controlling for data collection site and gender.

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
