# Peer review of "Children’s Internet Use Profiles in Relation to Behavioral Problems in Lithuania, Latvia, and Taiwan"

_ijerph, 2020, doi:10.3390/ijerph17228490_

Round 1

Reviewer 1 Report

This is a well-written report on the children´s internet use profiles in relation to behavioral problems among childrenstudents. The analysis conducted is clearly explained and the contribution to research is properly elaborated. The paper make a significant contribution to the research and have the potential to inform practice in important ways. Minor recommendations are suggested in order to enhance the usefulness of this manuscript.

Please, provide information about the Institutional approval of the study. Also, it is suggested to provide more details about the process of obtaining consent from children to participate in this study. Please, inculde the number of participants who refused to participate as well as missing data. Also, how were student selected? Inclusion and exclusion criteria.

Table 1 and 2 should be displayed together but excluding data of reliability (which are in text). Also, an only table of sociodemografic data and questionnaires should be included into results section instead of measures.

Finally, authors should be explain how this research have potential interest to informe educational practice for family and school.

Despite this limitations, the core of the article is important and relevant and I suggest the paper has potential after these changes.

Reviewer 2 Report

Review of the manuscript “Children’s Internet Use Profiles in Relation to Behavioral Problems in Lithuania, Latvia, and Taiwan”

The present projects assesses an important topic. The manuscript is well written. Differences in Internet use profiles according to gender and country are interesting questions. However, the article has several limitations. For example, the socio-economic status of the participants was either not assessed or not included in the analyses. Furthermore, the methods should be explained more clearly. They might also be improved, e.g., by including age (and SES) as covariates and by including all scales of the SDQ as outcomes measures.

Abstract:

  • Please mention the age range of participants in the abstract
  • Please add some more information on what “low, average, low-high, and high” mean. The reader should be able to understand how the three dimensions time, diversity, and knowledge are reflected in these classes.

Introduction:

  • When citing other studies, please include information on the age range of the children.
  • Line 123f (“For example,…”): This is no sentence.
  • Please include more information on why you have chosen these three countries already in the Introduction. You only mention the comparability of media use in these countries. Which DIFFERENECES lead you to choose these countries and which hypotheses did you have regarding media use classes?
  • In general, the introduction is very long. You may shorten (especially on page 2).

Methods

  • In Table 2, I suggest to write Cronbach’s alpha instead of Reliability.
  • When reading how the variable “diversity” was created, I realized that it reflects “frequency” rather than “diversity”. For example, a child that uses Internet for only two purposes, but each of these for several times per day (choosing option “5”), would reach a higher score than a child performing five activities only sometimes (e.g., choosing option “2” for each activity). However, the internet use of the second child would be more diverse than the internet use of the first. I, therefore, suggest to replace “diversity” by “frequency”. Another possibility would be to look, e.g., at how many of the 12 activities were performed at least xx times/week (I don’t know the exact answer options). This, indeed, would be a measure of diversity.
  • Please include associations with all scales of the SDQ (emotional, peer problems, hyperactivity/inattention, conduct, prosocial). For example, several studies found associations between media use and symptoms of hyperactivity/inattention in (young) children.
  • Why didn’t you assess age (in addition to site and gender) as covariate? I think Internet use changes between 8 and 10 years.
  • I fear I did not understand all the methods applied. Can you provide some more details on mixture models? Is it comparable with a regression?
  • Did you assess the socioeconomic status of the participating families? SES is an important covariate. You don’t mention SES in the manuscript. You should, at least, include it as a large limitation. All associations between Internet use and behavioral problems might be explained by differences in SES…

Results

  • I suggest clearer terms than just low, average, low-high, and high. The classes should reflect the content (knowledge, diversity/frequency, time). Especially the term low-high is problematic (what is low, what is high?)
  • For the analyses presented on page 9, was class (reference = low) the dependent measure and either site of gender the independent measure? In that case, didn’t you compare the probability of belonging to the other class (e.g., class “high”) in girls versus boys or in Lithuania versus Taiwan versus Latvia? If yes, I would expect, for example, in line 328, “… children from Taiwan have a lower probability of being in the High or Average Class compared to children from Lithuania or Latvia (I don’t know what was the reference here). In line 351f, I would expect “… that boys, compared to girls, had a higher risk of belonging to the high class (etc.)”. However, as mentioned before, I am not sure if I understood the methods you applied…
  • Lines 364-366 are not relevant in the Results section (it is already stated in the Introduction). Please remove.
  • Line 376: Chi-squared test? I thought, a chi-squared test is only suitable for categorical measures?
  • Were the results presented in Table 7 adjusted for site, gender (and age)?
  • Paragraph starting in line 396: This information should be included when describing the classes, but not after having described analyses on associations with behavioral difficulties.

Discussion

  • 442f: I suggest to remove these two sentences. They repeat the information provided in the previous sentence
  • I suggest to add some (short) information on differences in behavioral difficulties according to site (and gender).
  • Lines 491ff: Could reporting bias (social desirability) be an explanation for the unexpected finding of fewer children from Taiwan in the high class?

Round 2

Reviewer 2 Report

The current version of the manuscript reflects a clear improvement compared to the first version. The authors addressed all of my comments. I only have some minor comments:

  1. As you used all scales of the SDQ (as I appreciate), please also write this in the text. At some places, you still wrote “emotional and conduct problems”. I suggest writing “behavioral difficulties” instead, as behavioral difficulties refer to all scales (emotional problems, peer relationship problems, conduct problems, hyperactivity/inattention, low prosocial behavior).
  2. In the results section (especially lines 430 and 451), you should also indicate the significant association with hyperactivity (not only with conduct problems).
  3. In Table 7, I suggest to remove the letters in the title (“(a) Emotional problems, (b) hyperactivity/inattention …”). Readers might misinterpret them as relating to the letters in the table (which, however, indicate differences between classes).
